# Ubiquitin Ligase Nrdp1 Controls Autophagy-Associated Acrosome Biogenesis and Mitochondrial Arrangement during Spermiogenesis

**DOI:** 10.3390/cells12182211

**Published:** 2023-09-05

**Authors:** Zi-Yu Luo, Tian-Xia Jiang, Tao Zhang, Ping Xu, Xiao-Bo Qiu

**Affiliations:** 1Ministry of Education Key Laboratory of Cell Proliferation & Regulation Biology, College of Life Sciences, Beijing Normal University, 19 Xinjiekouwai Avenue, Beijing 100875, China; 201111201017@mail.bnu.edu.cn (Z.-Y.L.); jiangtx@bnu.edu.cn (T.-X.J.); 2State Key Laboratory of Proteomics, Beijing Proteome Research Center, Institute of Lifeomics, 38 Science Park Road, Beijing 102206, China; zhangtao120567648@163.com

**Keywords:** ubiquitin ligase, Nrdp1, Parkin, SIP/CacyBP, autophagy, acrosome, mitochondrium, mitophagy, spermiogenesis

## Abstract

Autophagy is critical to acrosome biogenesis and mitochondrial quality control, but the underlying mechanisms remain unclear. The ubiquitin ligase Nrdp1/RNF41 promotes ubiquitination of the mitophagy-associated Parkin and interacts with the pro-autophagic protein SIP/CacyBP. Here, we report that global deletion of Nrdp1 leads to formation of the round-headed sperm and male infertility by disrupting autophagy. Quantitative proteome analyses demonstrated that the expression of many proteins associated with mitochondria, lysosomes, and acrosomes was dysregulated in either spermatids or sperm of the Nrdp1-deficient mice. Deletion of Nrdp1 increased the levels of Parkin but decreased the levels of SIP, the mitochondrial fission protein Drp1 and the mitochondrial protein Tim23 in sperm, accompanied by the inhibition of autophagy, the impairment of acrosome biogenesis and the disruption of mitochondrial arrangement in sperm. Thus, our results uncover an essential role of Nrdp1 in spermiogenesis and male fertility by promoting autophagy, providing important clues to cope with the related male reproductive diseases.

## 1. Introduction

Spermiogenesis is the final phase of spermatogenesis characterized by condensation of DNA, replacement of nucleosomal histones with protamines, development of acrosome, and formation of the sperm tail connected with a mitochondria-rich midpiece [1,2]. The acrosome, located in the anterior part of the sperm nucleus, is a membranous lysosome-related organelle (LRO) [3,4]. Acrosome contains proteolytic enzymes that help sperm to enter an egg and is indispensable for fertilization [5]. Macroautophagy (hereafter referred to as autophagy) is an evolutionarily conserved process that degrades cellular proteins and damaged or excessive organelles. In this process, the double-membraned autophagosome fuses with lysosomes to form autolysosomes, where autophagic cargoes are degraded [6,7]. Formation of the autophagosome and selective recruitment of certain cargoes require LC3-II, whose precursor can be cleaved by ATG4 to form cytoplasmic LC3-I. LC3-I is then conjugated to phosphatidylethanolamine to form LC3-II on the phagophore membrane [8,9]. Mitophagy is an autophagic process to remove damaged or excessive mitochondria with induction of autophagy and recruitment of mitochondria [10]. Autophagy is critical to acrosome biogenesis [11], while mitophagy is important for mitochondrial organization in sperm [12]. However, how these processes are executed and regulated remains unclear.

The ubiquitin–proteasome pathway degrades most cellular proteins, and its dysregulation is involved in various diseases, including cancer and neurodegenerative diseases [13]. Inhibitors of the proteasome were wildly used in the clinical treatment of multiple myeloma and mantle cell lymphoma [14]. The ubiquitin ligase Nrdp1/RNF41 regulates cell growth, apoptosis, and autophagy by promoting ubiquitination of multiple important substrates, including the epidermal growth factor receptor family member ErbB3/HER3 [15], the pro-autophagic factor SIP/CacyBP [16], the dual inhibitor of apoptosis and autophagy BIRC6/BRUCE [16,17] and the mitophagy-associated ubiquitin ligase Parkin [18]. Parkin can maintain mitochondrial quality by promoting mitophagy [19]. Drp1 plays an important role in Parkin-mediated mitophagy by promoting mitochondrial fission [20,21]. Recently, we showed that Nrdp1 promotes monoubiquitination of SIP/CacyBP, enhances its association with Rab8, BIRC6, and LC3-I in the trans-Golgi network, and thus facilitates autophagosome formation and autophagic degradation of BIRC6 [16]. Nrdp1 is highly expressed in the testis, pancreas, brain, and muscles [15,22]. To understand the physiological roles of Nrdp1, we generated the mice with global deletion of Nrdp1 and found that the Nrdp1-deficient male mice are infertile with impaired spermiogenesis. Notably, our results suggest that Nrdp1 controls the mitochondrial arrangement in sperm and acrosome biogenesis by promoting autophagy.

## 2. Materials and Methods

### 2.1. Mice

Construction of the Nrdp1-deficient mice was achieved by Biocytogen Pharmaceuticals Co., Ltd. (Beijing, China). Mice were kept in the Institute of Brain and Cognitive Sciences, Beijing Normal University using standard humane animal husbandry protocols. The animals’ care was in accordance with institutional guidelines. Unless stated elsewhere, mice were 6 per group with age- and sex-matched in each experiment. Sample size was based on empirical data from pilot experiments. No additional randomization or blinding was used to allocate experimental groups. For genotyping, DNA was extracted from the tip of the tail and analyzed by PCR with the primers: wild-type: 5′-CCACACGCAGCCCTCGTACTC (forward), 5′-AGGTCCAGGGCTCACAATCAAGG (reverse); Nrdp1-deficient: 5′-CCACACGCAGCCCTCGTACTC (forward), and 5′-CCAGCACCCTGCCA CGGATT (reverse).

### 2.2. Antibody Information

The antibody for Sp56 was kindly gifted from Prof. Fei Gao. Other antibodies were purchased according to the following information: Afaf (Abcam, Cambridge, UK, ab121470); Drp1 (BD bioscience, Franklin Lakes, NJ, USA, 611113); GM130 (Abcam, ab52649); LAMP1 (Sigma-Aldrich, St. Louis, MO, USA, L1418); LC3B (Sigma-Aldrich, L7543); Nrdp1 (Santa Cruz Biotechnology, Dallas, TX, USA, sc-365622); OPTN (Proteintech, Rosemont, IL, USA, 10837-1-A); Parkin (Sigma-Aldrich, P6248); p62 (Cell Signaling Technology, Danvers, MA, USA, 88588); SIP/CacyBP (Santa Cruz, sc-166455); Sox9 (Abcam, ab185966); Tim23 (BD bioscience; 611222); VAMP8 (Abcam, ab76021); Ubiquitin (Cell Signaling Technology, 3936S); and β-actin (Sigma-Aldrich, A5441). Peroxidase-conjugated anti-mouse IgG (ZSGB-BIO, Beijing, China, ZB-5305, 1:5000), anti-rat IgG (ZSGB-BIO, ZB-2307, 1:4000), or anti-rabbit IgG (ZSGB-BIO, ZB-5301, 1:3000) was used as the secondary antibody. The protein bands were visualized using the ECL detection system (Millipore, Burlington, MA, USA).

### 2.3. Immunoblotting

Unless stated elsewhere, testes and sperm were homogenized in the buffer containing 20 mM Tris-HCl (pH 8.0), 100 mM KCl, 1 mM EDTA, 1 mM EGTA, 1% Triton X-100, 2.5 mM sodium pyrophosphate, 1 mM β-sodium glycerophosphate and protease inhibitors, sonicated twice at 200 W for 10 s each, and then cleared by centrifugation. For the detection of sperm ubiquitination, we used 1.2× SDS sample buffer as a lysis buffer to denature deubiquitinating enzymes and the ubiquitinated proteins. Proteins were separated by SDS-PAGE. After proteins were transferred to a PVDF membrane (Millipore), the blot was incubated with the indicated primary antibodies. The secondary antibody was goat against rabbit or mouse IgGs conjugated to horseradish peroxidase (HRP).

### 2.4. Tissue Collection and Immunostaining

Testes were fixed in 4% paraformaldehyde (PFA) at 4 °C overnight, dehydrated, embedded in paraffin, and sectioned at 5 μm. The sections were de-paraffinized, rehydrated, and followed by antigen retrieval in 10 mM of the sodium citrate buffer. Then, sections were blocked with goat serum in 0.3% triton X-100 and incubated with primary antibodies. In the case of a sperm smear, the section can be blocked directly and incubated with primary antibodies. The secondary antibody was goat against rabbit or mouse IgGs conjugated to FITC, and DAPI was used to stain the nuclei of cells.

### 2.5. Hematoxylin and Eosin (H&E) Staining

Mouse testis and epididymis were excised and fixed in 4% paraformaldehyde overnight. Sectioned at 5 μm and stained by hematoxylin and eosin. The cytoplasm was stained by eosin (red), and the nucleus was stained by hematoxylin (blue).

### 2.6. Separation of Testis Haploid Cells by Flow Cytometry

Testis dissociation and cell collection were based on a recently described method [23]. Testicles were isolated, de-capsulated, and incubated in the solution containing 1 mg/mL of collagenase, 0.5 mg/mL DNase I, and 1 mg/mL hyaluronidase. The tube was incubated for 10 min at 37 °C, and then seminiferous tubules were incubated in pre-heated solution with 1 mg/mL of collagenase, 0.5 mg/mL DNase I, and 1 mg/mL trypsin for 10 min at 37 °C. The tubules were gently pipetted up and down. The suspension was passed through a 100 μm nylon cell strainer and washed with 1× PBS. The cells were then incubated with Hoechst33342. The cells were finally sorted by a flow cytometer (BD FACSAria™ III) and analyzed using BD FACSDiva™. Haploid cells were collected based on Hoechst33342 fluorescence intensity.

### 2.7. Mass Spectrometry

Protein samples were collected from sperm and testis haploid cells of heterozygous deletion of Nrdp1 (*Nrdp1*^+/−^) and Nrdp1-deficient (*Nrdp1*^−/−^) mice, and analyzed for differentially expressed proteins by mass spectrometry at Beijing Proteome Research Center using the methods described previously [24,25]. Briefly, the protein samples were assayed for quality. Equal amounts of each group were taken for protein alkylation and trypsin digestion to obtain peptides. Then, the peptides were labeled with 8-channel Tandem Mass Tag (TMT) reagents, equilibrated, desalted, and cleaned up for the Liquid Chromatograph–Mass Spectrometry/Mass Spectrometry (LC-MS/MS) analysis and quantitative studies. The fold changes between the two groups were calculated by dividing the average abundance values of each protein in KO samples by the ones in the control group. The threshold value of proteins upregulated was 1.5, and the downregulated value was 0.67.

### 2.8. Pathway and Process Enrichment Analyses

For GO term pathway and process enrichment, the analysis has been separately carried out using the “micro-bioinformatics” Gene Annotation and Analysis Resource (https://www.bioinformatics.com.cn/plot_basic_gopathway_enrichment_bubbleplot_081, accessed on 29 July 2023). We choose our differentially expressed proteins in *Nrdp1*^−/−^ testis haploid cells and sperm and obtain a bubble plot.

### 2.9. Quantification and Statistical Analysis

Unless stated elsewhere, significance levels for comparisons between two groups were determined by a two-tailed unpaired *t*-test, mean and s.e.m. (* *p* < 0.05, ** *p* < 0.01, and *** *p* < 0.001), and normal distribution. All of the images were chosen blindingly and randomly and quantitated by image J (V1.8.0).

## 3. Results

### 3.1. Global Deletion of Nrdp1 Leads to Male Infertility

To investigate the physiological role of Nrdp1, we generated the mutant mice with global deletion of the Nrdp1 gene by using Clustered Regularly Interspaced Short Palindromic Repeats (CRISPR)–Cas9 technology (Figure 1A). Nrdp1 has 7 exons with the translated region stretching across exons 3–7. In this study, the cut sites by Cas9 are located on both sides of exon 4 of Nrdp1, resulting in the cleavage of the main region of Nrdp1 mRNA and loss of Nrdp1 protein. PCR analyses of genomic DNA indicated that the whole exon 4 has been cut off (Appendix A). Further confirmation of the successful deletion of Nrdp1 was carried out by immunoblotting in different organs, such as the testis, kidney, and spleen, and the results showed that the Nrdp1-deficient mice (Nrdp1^−/−^) have been constructed successfully (Figure 1B). Homozygous deletion of Nrdp1 caused male infertility, and none of the females that bred with Nrdp1^−/−^ males were pregnant. But Nrdp1^−/−^ females could produce offspring normally with similar-sized litters as their wild-type or heterozygous littermates (Figure 1C). Intriguingly, there were no significant differences in the volume and weight of testicles (Figure 1D). In addition, no obvious defects were found in spermatogonia, spermatocytes, round spermatids, and elongated spermatids within the seminiferous tubules of Nrdp1^−/−^ mice as examined by hematoxylin–eosin (HE) staining (Figure 1E). No difference in the Nrdp1-deficient Sertoli cells in testes as marked by Sox9 was observed (Appendix A). On the other hand, no defects were observed in the ovaries, uterus, and follicular cells in the Nrdp1-deficient female mice (Figure 1F,G). These results suggest that Nrdp1 deficiency leads to male infertility probably due to the impaired spermatogenesis after spermatid elongation.

### 3.2. Nrdp1 Deficiency Causes Round-Headed Sperm with Disrupted Mitochondrial Arrangement

We next analyzed sperms in the mouse epididymis. The heads of almost all sperm in the epididymis of Nrdp1^−/−^ mice were round, unlike the sickle shape in the wild-type mice (Appendix A). But the sperm count was not significantly affected in the Nrdp1^−/−^ mice (Appendix A). In addition, more than 50% of Nrdp1^−/−^ sperm had multiple tails and swollen middle pieces of the tail (Figure 2A) with a sharp decrease in sperm motility (Figure 2B and Appendix A). Accordingly, mitochondria failed to align in the tail but accumulated in the head in Nrdp1^−/−^ sperm (Figure 2C). Transmission electron microscopy with epididymis sections further confirmed mitochondrial defects with the abnormal structure of mitochondria in Nrdp1^−/−^ sperm. The structure of mitochondria in the sperm head was mostly abnormal in Nrdp1^−/−^ samples (Figure 2D). Thus, Nrdp1 deficiency causes round-headed sperm and disrupts mitochondrial arrangement in sperm, leading to the sharply reduced motility of sperm.

Acrosome formation is a multiple-step process, including Golgi, cap, acrosome, and maturation phases [5]. Immunostaining assay with the acrosome membrane protein Afaf [26] showed that the shape of the sperm head was severely disrupted in the Nrdp1^−/−^ mice in comparison to the regular sickle shape in the wild-type (Figure 3A). Transmission electron microscopy analyses with epididymis sections further showed nuclear and acrosomal defects in Nrdp1^−/−^ sperm. Acrosomes were absent or malformed in most Nrdp1^−/−^ sperm. At the same time, their nuclei were smaller (Figure 3B). To figure out which stages were impaired during acrosome formation, the Golgi phase was visualized by an antibody against the Golgi matrix protein GM130, while the cap, acrosome, and maturation phases were marked with the acrosome-specific protein Afaf. In the Golgi phase, there was no difference in Golgi or proacrosomal vesicles between Nrdp1^+/+^ and Nrdp1^−/−^ spermatids. In the cap phase, the length of acrosome anchoring nuclei in Nrdp1^−/−^ spermatids was much shorter than that in the wild-type. The disordered distribution of acrosomes occurred in both acrosome and maturation phases in Nrdp1^−/−^ spermatids (Figure 4A). Under a transmission electron microscope, the acrosomes in the Golgi phase were larger and slightly invaginated toward the nucleus in Nrdp1^−/−^ spermatids (Figure 4B). In the cap phase, the acrosomes of Nrdp1^+/+^ and Nrdp1^+/−^ sperm were intact and anchoring tightly to the nucleus, with a length close to half of the nucleus perimeter, while the acrosomes of Nrdp1^−/−^ spermatids were shorter with a length of about only one-third of the nucleus perimeter (Figure 4B). In the acrosome phase, the manchette microtubule is the major factor shaping nuclei of the acrosome during the early elongation phase of spermiogenesis [27]. However, the structure of the manchette was dramatically disturbed in the acrosome phase of Nrdp1^−/−^ spermatids. In the mature phase, the acrosome was not located anteriorly, and small vacuoles appeared in Nrdp1^−/−^ spermatids (Figure 4B). Thus, Nrdp1 deletion markedly impairs the development of acrosomes in the late phases of acrosome formation.

### 3.3. Nrdp1 Deficiency Disrupts Expressions of Lysosomal and Mitochondrial Proteins in Spermatids and Sperm

To explore the mechanisms by which Nrdp1 regulates spermiogenesis, we performed the TMT-labeled quantitative proteome analysis. Haploid spermatids and immature sperm in mouse testes were stained in nuclei with Hoechst33342 and sorted by a fluorescence-activated cell sorter (FACS), while the late-stage or mature sperm were isolated from mouse epididymis (Figure 5A,B). There were 440 upregulated and 201 downregulated differentially expressed proteins in the Nrdp1-deficient (Nrdp1^−/−^) testis haploid cells in comparison with the control samples from the mice with the heterozygous deletion of Nrdp1 (Nrdp1^+/−^) (Figure 5C,D), which showed no defects in spermiogenesis. In the Nrdp1-deficient epididymis sperm, there were 374 upregulated and 113 downregulated differentially expressed proteins (Figure 5E,F). Eighteen proteins, which were mostly transmembrane proteins, were co-upregulated in the Nrdp1-deficient testis and epididymis samples (Figure 5G and Appendix A). Three proteins (Cyp2b10, Fam96b, and Tmem263) were upregulated in testes, but downregulated in epididymis samples (Figure 5H and Appendix A). Ints12 was the only one downregulated in both testis and epididymis samples (Figure 5I). Apparently, the deletion of Nrdp1 led to more proteins being upregulated in both testis and epididymis samples, consistent with the role of Nrdp1 in ubiquitination, which regulates protein degradation through both proteasomes and lysosomes [28,29,30]. These results also suggest that Nrdp1 might play relatively distinct roles in immature haploid cells, including round spermatids, elongated spermatids, and immature sperm in the testis, and the mature sperm characterized with the well-oriented mitochondria and acrosome in the epididymis, supporting a critical role of Nrdp1 in sperm maturation.

Lysosomal proteins were found highly upregulated in the Nrdp1^−/−^ testis haploid cells using KEGG pathway analyses (Figure 6A). In addition, sixteen mitochondrial function-related proteins were upregulated in the Nrdp1-deficient haploid cells from the testes (Appendix A). Contrary to the results from testis samples, the deletion of Nrdp1 caused the lysosome pathway to downregulate in the Nrdp1^−/−^ sperm from the epididymis (Figure 6B). The GO term analyses showed that proteins in mitochondria and transport vesicles were ranked at the top as upregulated proteins in the Nrdp1-deficient epididymis sperm (Figure 6C and Appendix A). Among them, the mitochondrial autophagy-related Parkin (Park2) [31], the lysosomal membrane protein LAMP1 [32], and 35 proteins involved in mitochondrial ATP synthesis or NADH dehydrogenase were all upregulated (Appendix A). In contrast, multiple acrosome-associated proteins were downregulated in the Nrdp1-deficient haploid cells from testes and sperm in the epididymis (Figure 6D and Appendix A; Appendix A).

Immunoblotting analyses demonstrated that deletion of Nrdp1 decreases the levels of the autophagic marker LC3-II, the lysosome membrane protein LAMP1, the vesicle transfer and fusion protein VAMP8 [33], and the autophagic cargo receptor OPTN [34], while increasing the levels of the cargo receptor as well as an autophagic substrate p62 [35] in mouse testes (Figure 7A), suggesting that autophagy is generally inhibited in the Nrdp1-deficient testes. Immunoblotting analyses demonstrated that Nrdp1 deletion obviously increased the levels of Parkin, but decreased the levels of LC3, the mitochondrial fission protein Drp1, the autophagy-promoting protein SIP, and mitochondrial inner membrane protein Tim23 [36] in sperm from the epididymis (Figure 7B), indicating that autophagy is also inhibited in the Nrdp1-deficient sperm in the epididymis. The levels of the ubiquitinated proteins at high molecular weights in Nrdp1-deficient sperm are slightly higher than those in the wild-type (Figure 7C). But, several ubiquitinated bands at low molecular weights almost disappeared in the Nrdp1-deficient sperm. The sizes of these downregulated ubiquitinated proteins are above that of non-ubiquitinated Parkin (52 kDa) (Figure 7C), consistent with the role of Nrdp1 as a ubiquitin ligase of Parkin [18]. Immunoblotting analyses also showed that the levels of the acrosomal marker Sp56 [37] decreased in the Nrdp1^−/−^ sperm from the epididymis (Figure 7D), further supporting that the autophagy-dependent formation of acrosome requires Nrdp1. In addition, the deletion of Nrdp1 shifted the distribution of mitochondrial Parkin and Drp1 from the tail to the head of sperm, accompanying the shifting of mitochondrial localization (Figure 7E,F). These results suggest that Nrdp1 plays a critical role in mitochondrial localization and function during spermiogenesis at least partially by promoting Parkin ubiquitination. Thus, Nrdp1 is required for programmed autophagy, acrosome biogenesis, and proper mitochondrial function during spermiogenesis.

## 4. Discussion

Autophagy plays an important role in sperm acrosome formation [38,39]. Deficiency of the autophagic protein ATG7 in male mouse germ cells leads to the production of round-headed sperm with abnormal acrosome development [11]. The deletion of Sirt1, which deacetylates ATG7 and LC3, also leads to similar phenotypes in mouse sperm [40]. The phenotypes of Nrdp1 deletion were similar to those of the germ cell-specific knockout of Atg7, which were apparently due to the inhibition of autophagy, because the levels of the autophagy-promoting protein SIP, the autophagy marker LC3-II, and the mitochondrial inner membrane protein Tim23 were all reduced in sperm by Nrdp1 deletion. Parkin mutations lead to an autosomal recessive form of Parkinson’s disease due to its role in regulating mitophagy for mitochondrial quality control [41]. Nrdp1 promotes Parkin ubiquitination, and regulates the intracellular activity of Parkin [18], the latter of which promotes ubiquitination and degradation of the mitochondrial fission protein Drp1 [42]. While the levels of the other Nrdp1 substrates in sperm, including ErbB3 [15] and BIRC6/BRUCE [17], were not elevated by Nrdp1 deletion (Appendix A [43,44]), this study shows that deletion of Nrdp1 increased the levels of Parkin, but reduced the levels of Drp1 in sperm, suggesting that the suppressed ubiquitination of Parkin at least partially contributes to the abnormal arrangement of mitochondria in sperm, the impaired sperm motility, and male infertility. Among the substrates of Nrdp1, Parkin is the only one upregulated in Nrdp1-deficient mice sperm. Thus, Parkin might be the main substrate of Nrdp1 spermiogenesis through controlling autophagy and mitochondrial quality. Due to the lack of an efficient in vitro sperm culture system, this study did not provide any rescuing data by placing the Nrdp1 gene back into the Nrdp1-deficient sperm. But our quantitative proteome analyses showed that expression of the proteins associated with mitochondria, lysosomes and acrosomes is largely dysregulated in either spermatids or sperm of the Nrdp1-deficient mice, validating that the autophagic pathway is indeed affected by deletion of Nrdp1. These results are consistent with the known roles of Nrdp1 in enhancing the association of the pro-autophagy protein SIP/CacyBP with the trans-Golgi network to promote autophagosome formation and promoting the ubiquitin-mediated degradation of the mitophagy-regulating protein Parkin [16,18]. Nrdp1 exists in both the trans-Golgi network and recycling endosomes, and SIP probably regulates the Nrdp1-mediated translocation of BRUCE and LC3-I from the trans-Golgi network to the recycling endosome [16]. Thus, Nrdp1 should form a complex with BRUCE and LC3-I, which is regulated by SIP. But how this complex is regulated for degrading Parkin during spermiogenesis remains to be determined in the future. The sperm phenotypes of the Nrdp1-deficient mice are very similar to those of globozoospermia, a rare reproductive disease characterized by acrosome absence, lack of post-acrosomal sheaths, and abnormal mitochondrial sheaths [45,46].

## 5. Conclusions

We found that global deletion of Nrdp1 in mice disrupts mitochondrial arrangement in sperm. During spermiogenesis, Nrdp1 deletion disrupts acrosome biogenesis and results in round-headed sperm. Nrdp1 controls the Parkin levels and promotes autophagy in sperm. In conclusion, we have found that the ubiquitin ligase Nrdp1 is essential for spermiogenesis and male fertility by regulating autophagy. These results may provide new clues to cope with the related male reproductive diseases.

## Figures and Tables

**Figure 1 cells-12-02211-f001:**
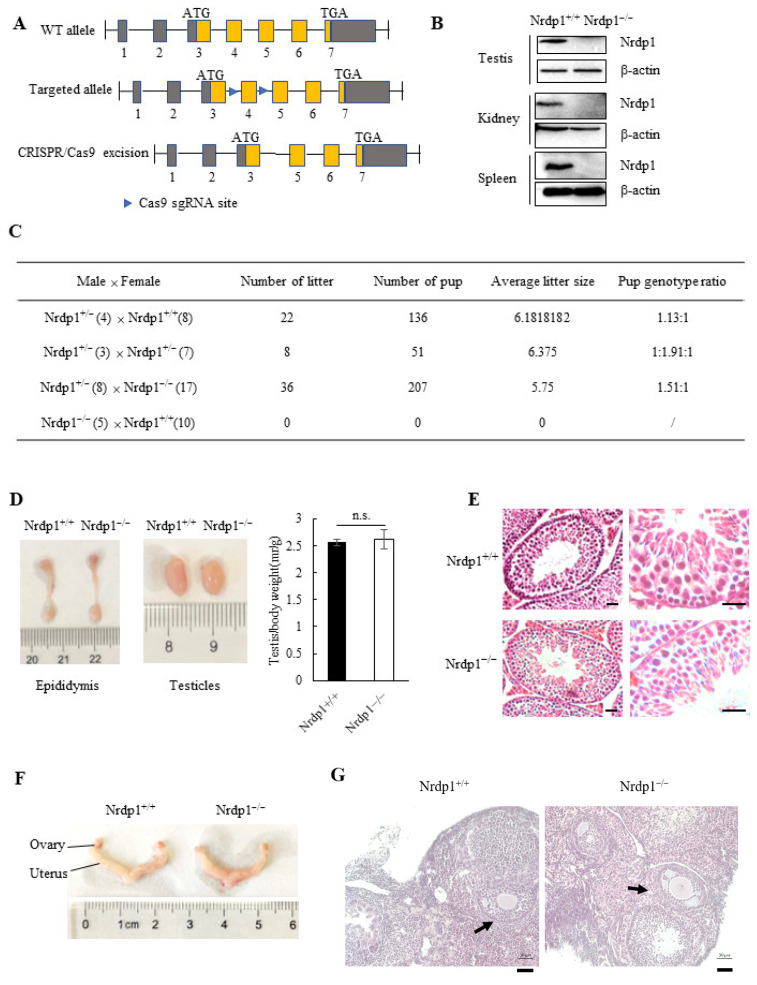
Deletion of Nrdp1 leads to male infertility in mice. (**A**) Schematic diagram of Nrdp1 deletion in mice. The sgRNAs were designed to target two flanks between the exon 4 of *Nrdp1* gene. The sequence alignment indicated a 1037 bp loss (including the whole exon 4) that led to the frameshift mutation. WT, the wild-type. (**B**) Immunoblotting for Nrdp1 protein in the wild-type (*Nrdp1*^+/+^) and the Nrdp1-deficient (*Nrdp1*^−/−^) male tissues, including testes, spleens, and kidneys. β-actin was used as the loading control. (**C**) Fertility test of *Nrdp1*^−/−^ males and females. Adult males of *Nrdp1*^+/−^ or *Nrdp1*^−/−^ genetic background were mated to *Nrdp1*^+/+^, *Nrdp1*^+/−^ or *Nrdp1*^−/−^ females. The number of mating mice was indicated in parentheses. The number of litters and pups was recorded for more than two months. (**D**) The size of the testes and epididymis from *Nrdp1*^+/+^ and *Nrdp1*^−/−^ male mice. *n* = 4. (**E**) The morphology of the seminiferous tubules from *Nrdp1*^+/+^ and *Nrdp1*^−/−^ mice was examined by H&E staining. Scale bar = 50 μm. (**F**) The morphology of the ovary and uterus from *Nrdp1*^+/+^ and *Nrdp1*^−/−^ mice was shown. (**G**) The morphology of the follicular cells from *Nrdp1*^+/+^ and *Nrdp1*^−/−^ mice was examined by H&E staining. Black arrow indicating follicles. Scale bar = 50 μm. Data are presented as mean ± SEM. n.s. stands for not significant. Data are from four independent biological replicates of the same experiment (two-tailed unpaired *t*-test in (**D**)).

**Figure 2 cells-12-02211-f002:**
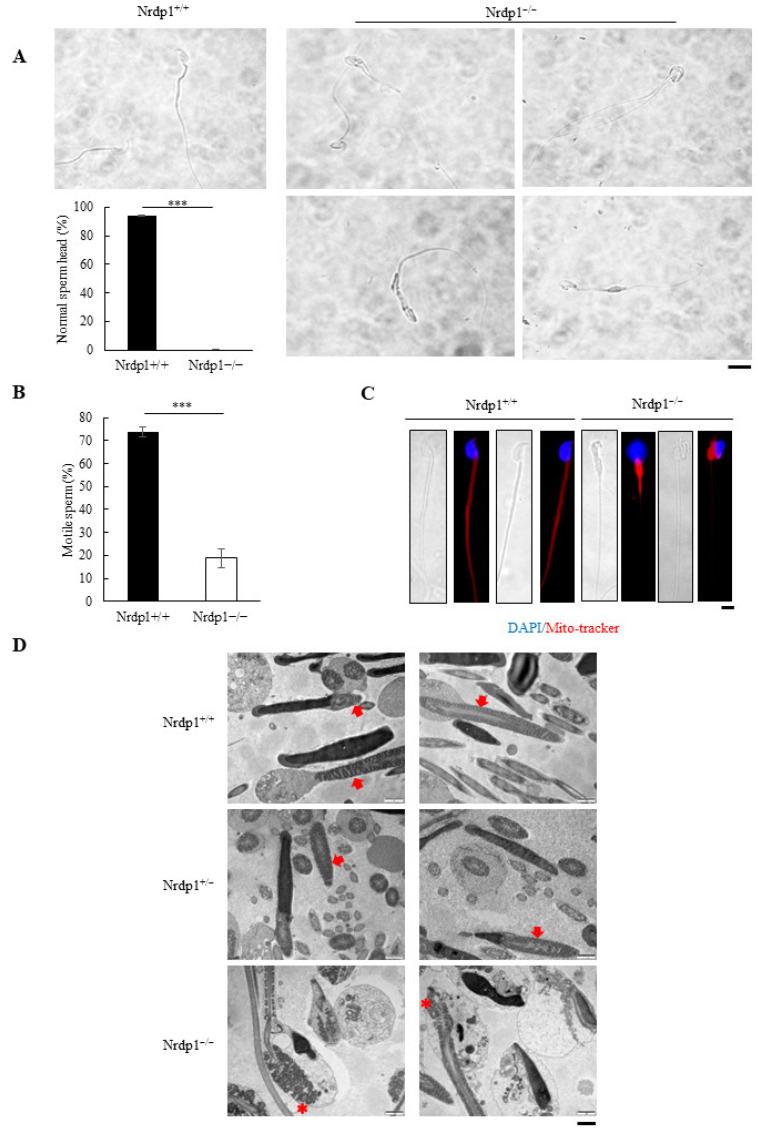
*Nrdp1*^−/−^ mice display structural defects in sperm. (**A**) Single sperm image for the morphology of *Nrdp1*^+/+^ (left panel) and *Nrdp1*^−/−^ sperm (right panel) derived from cauda epididymis. The percentage of normal or round-headed sperm between *Nrdp1*^+/+^ and *Nrdp1*^−/−^ mice was calculated. At least 10 random sections were counted. Scale bar = 10 μm. (**B**) The percentage of motile sperm between *Nrdp1*^+/+^ and *Nrdp1*^−/−^ mice. *n* = 4. (**C**) Fluorescence staining of single sperm using mito-tracker (red), which labeled mitochondrial sheath in the mid-piece of the sperm tail. The nuclei were stained by DAPI (blue). Scale bar = 10 μm.
(**D**) The ultrastructure of mitochondria (indicated by arrows in *Nrdp1*^+/+^, *Nrdp1*^+/−^ mice and asterisks in
*Nrdp1*^−/−^ mice) in sperm using TEM. Scale bar = 1 μm. Data are presented as mean ± SEM. *** *p* < 0.001. Data are from four independent biological replicates of the same experiment (two-tailed unpaired *t*-test in (**A**,**B**)).

**Figure 3 cells-12-02211-f003:**
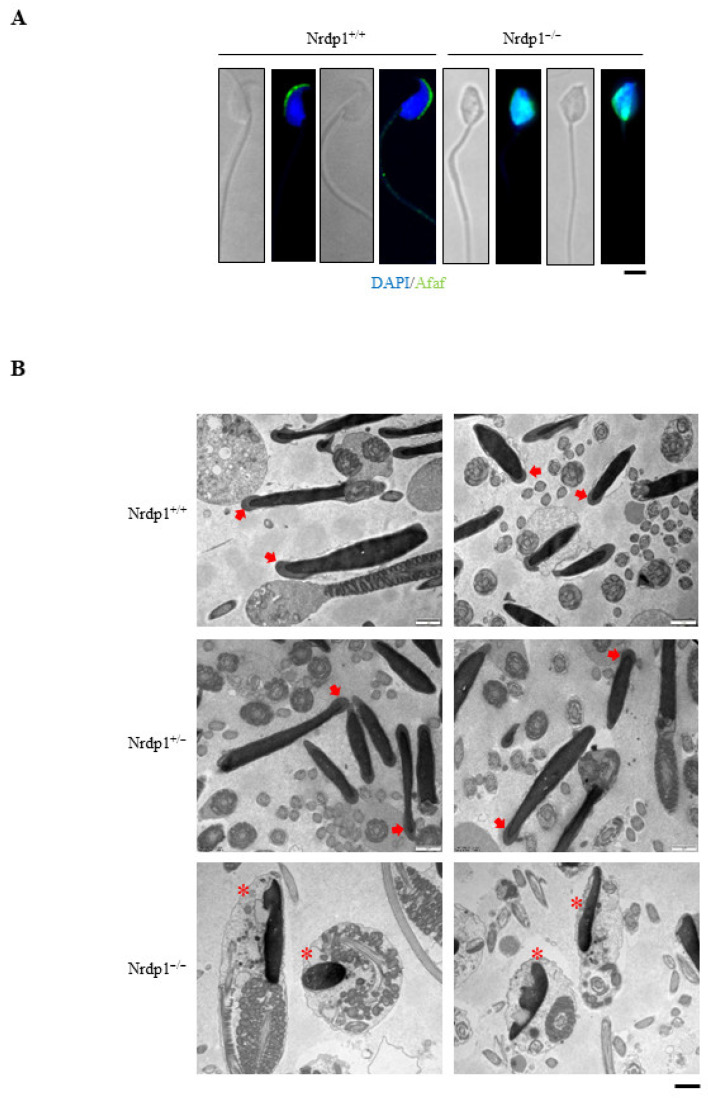
Acrosome structure is impaired in *Nrdp1*-deficient sperm. (**A**) Immunofluorescence staining of single sperm using an antibody against the acrosome-specific protein Afaf (green). The nuclei were stained by DAPI (blue). Scale bar = 10 μm. (**B**) The ultrastructure of acrosome (indicated by the arrow in *Nrdp1*^+/+^, *Nrdp1*^+/−^ mice and asterisks in *Nrdp1*^−/−^ mice) in sperm using TEM. Scale bar = 1 μm. Data are from three independent biological replicates of the same experiment.

**Figure 4 cells-12-02211-f004:**
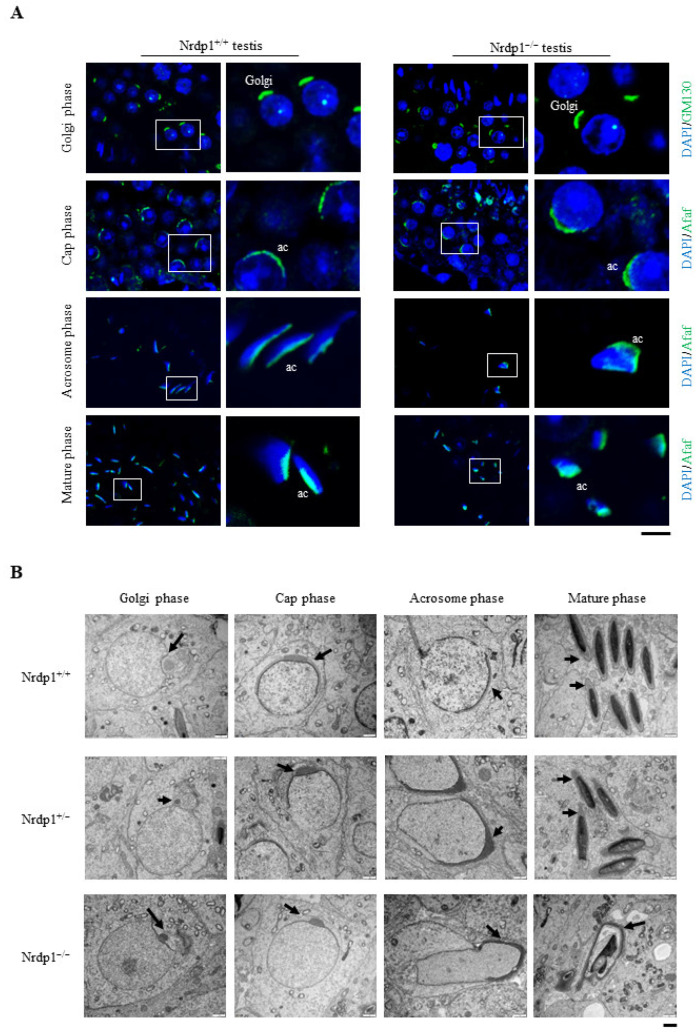
Acrosome formation is disrupted in *Nrdp1*-deficient mice. (**A**) Adult testis sections from *Nrdp1*^+/+^ and *Nrdp1*^−/−^ samples were visualized by the antibody against the Golgi marker GM130 (green). Acrosome membrane protein Afaf (green) was used to label developing acrosome, and DAPI was for nuclei (blue). In the cap phase, both *Nrdp1*^+/+^ and *Nrdp1*^−/−^ round spermatids display the characteristic acrosomal caps that covered the nuclei. In the acrosome phase, Afaf-labelled acrosome elongated along with cell nuclei in *Nrdp1*^+/+^ testis samples, whereas Afaf staining appeared disarranged in *Nrdp1*^−/−^ elongated spermatids. ac, acrosome. Scale bar = 10 μm. (**B**) The ultrastructure of the Golgi phase showed the proacrosomal granule (arrows) around the nucleus in *Nrdp1*^+/+^, *Nrdp1*^+/−^ and *Nrdp1*^−/−^ round spermatids. As in *Nrdp1*^+/+^ spermatids, acrosomes (arrows) extended along the nucleus in *Nrdp1*^−/−^ round spermatids during the cap phase. In the acrosome phase and the mature phase, irregularly shaped acrosomes (arrows) were shown in *Nrdp1*^−/−^ elongated spermatids. Scale bar = 1 μm. Data are from two independent biological replicates of the same experiment.

**Figure 5 cells-12-02211-f005:**
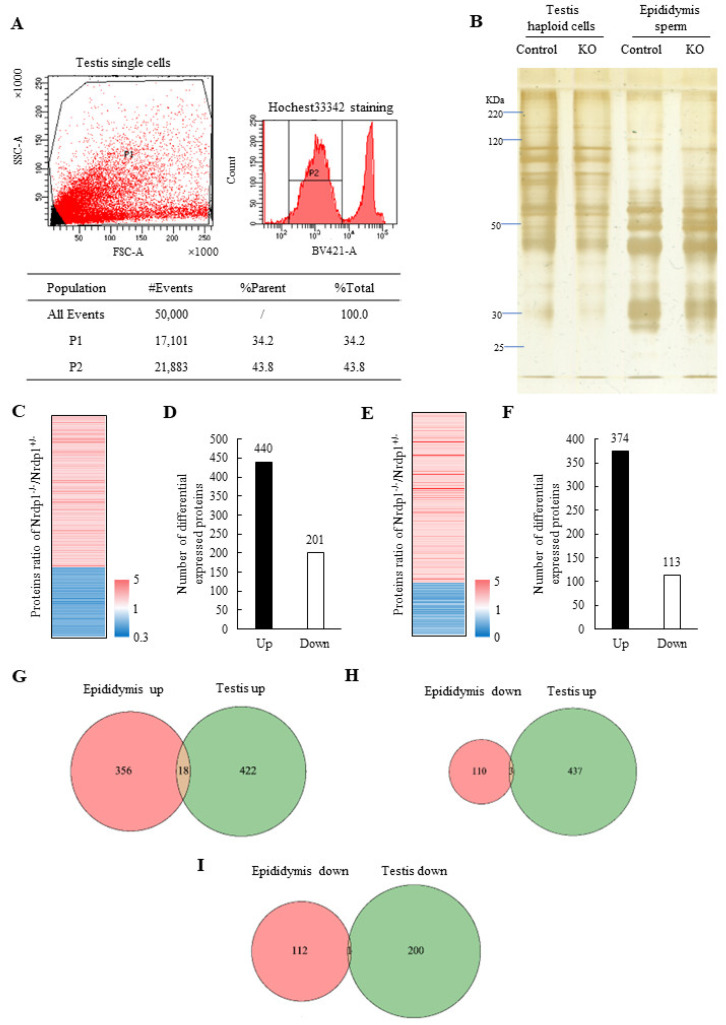
Quantitative proteome analysis of sperm and testis haploid cells in *Nrdp1*^+/−^ and *Nrdp1*^−/−^ mice. (**A**) FACS gating strategy for isolation of haploid cells using Hoechst33342 staining (P2 group). (**B**) Proteins from testis haploid cells and sperm of *Nrdp1*^+/−^ (control) and *Nrdp1*^−/−^ (KO) mice were analyzed prior to mass spectrometric detection by SDS-PAGE and silver staining. (**C**) Differential expressed proteins in testis haploid cells from *Nrdp1*^+/−^ and *Nrdp1*^−/−^ mice. (**D**) Up- and downregulated proteins in testis haploid cells from *Nrdp1*^+/−^ and *Nrdp1*^−/−^ mice. (**E**) Differential expressed proteins in sperm from *Nrdp1*^+/−^ and *Nrdp1*^−/−^ mice. (**F**) Up- and downregulated proteins in spermatozoa from *Nrdp1*^+/−^ and *Nrdp1*^−/−^ mice. (**G**) Co-upregulated proteins in both sperm and testis haploid cells from *Nrdp1*^−/−^ mice. (**H**) Upregulated proteins in *Nrdp1*^−/−^ mice testis haploid cells and downregulated in *Nrdp1*^−/−^ mice sperm. (**I**) Co-downregulated proteins in both sperm and testis haploid cells in *Nrdp1*^−/−^ mice.

**Figure 6 cells-12-02211-f006:**
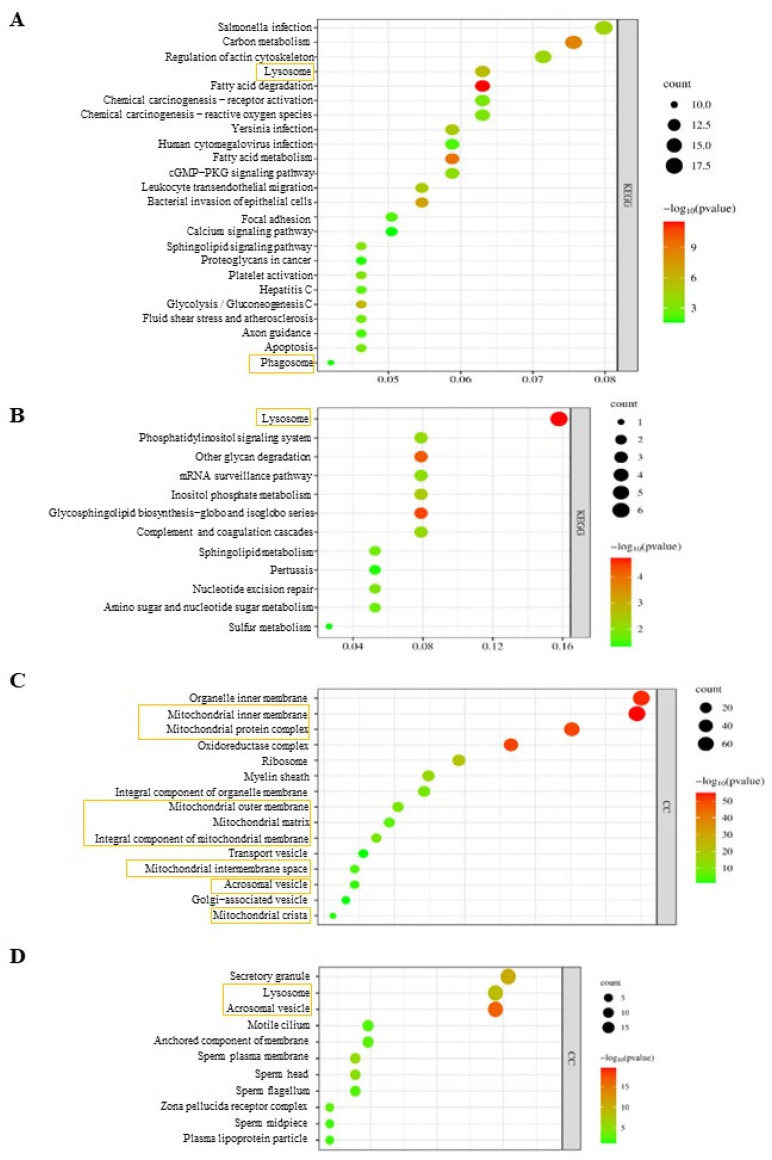
Pathway and enrichment analyses in *Nrdp1*^+/−^ and *Nrdp1*^−/−^ mouse haploid cells in testis or epididymis. (**A**) KEGG-pathway analysis of the proteins upregulated (440) in *Nrdp1*^−/−^ testis haploid cells, colored by *p*-values. (**B**) KEGG-pathway analysis of the proteins downregulated (113) in *Nrdp1*^−/−^ sperm from the epididymis, colored by *p*-values. (**C**) Cellular component enrichment GO term analysis of upregulated expressed proteins (374) in *Nrdp1*^−/−^ sperm, CC: cellular component, colored by *p*-values. (**D**) Cellular component enrichment GO term analysis of downregulated expressed proteins (113) in *Nrdp1*^−/−^ sperm, CC: cellular component, colored by *p*-values. The proceeds and cellular components in yellow boxes stand for which were associated with mitochondria, acrosome and autophagy.

**Figure 7 cells-12-02211-f007:**
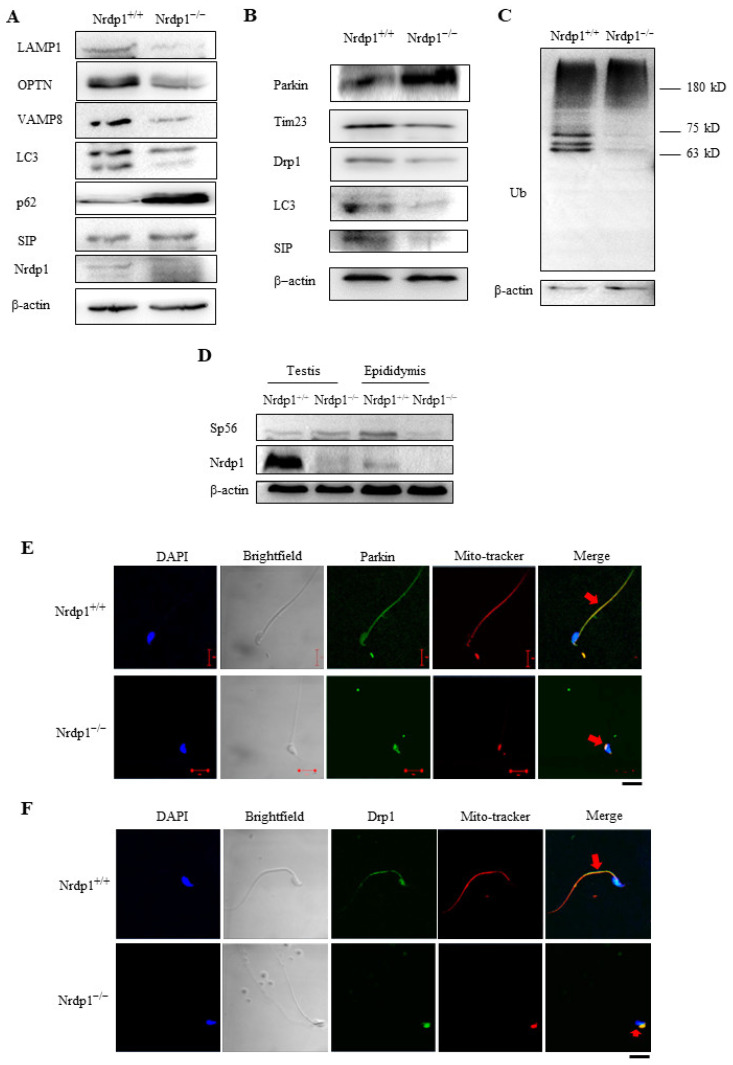
Nrdp1 deficiency disrupts expression or localization of lysosomal or mitochondrial proteins in spermatids and sperm. (**A**–**C**) Immunoblotting of the testis (**A**) or sperm (**B**,**C**) homogenates from *Nrdp1*^+/+^ and *Nrdp1*^−/−^ mice. β-actin was used as the loading control. (**D**) Immunoblotting of testis and epididymis homogenates from *Nrdp1*^+/+^ and *Nrdp1*^−/−^ mice. β-actin was used as the loading control. (**E**,**F**) Immunofluorescence analyses of sperm using antibodies against Parkin (**E**) or Drp1 (**F**) (green). Mitochondria were stained by MitoTracker (red). The nuclei were stained by DAPI (blue). The red arrows stand for colocolization of Parkin or Drp1 with mitochondria. Scale bar = 10 μm. Data are from three independent biological replicates of the same experiment.

## Data Availability

The data that support the findings of this study are available from the authors upon request, and requests for materials should be addressed to X.-B.Q. (xqiu@bnu.edu.cn).

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
