# Peer review of "Ubiquitin Ligase Nrdp1 Controls Autophagy-Associated Acrosome Biogenesis and Mitochondrial Arrangement during Spermiogenesis"

_cells, 2023, doi:10.3390/cells12182211_

Round 1

Reviewer 1 Report

ID:       Cells                                                                                                  MDPI                         

Title:    Ubiquitin ligase Nrdp1 controls autophagy-associated acrosome biogenesis and mitochondrial arrangement during spermiogenesis

Authors: Zi-Yu Luo, Tao Zhang, Ping Xu, Tian-Xia Jiang, Xiao-Bo Qiu,                      

In this manuscript the authors show the role of the ubiquitin ligase Nrdp1/RNF41 during spermiogenesis. It is known that Nrdp1 ubiquitinates Parkin and interacts with SIP/CacyBP. In the manuscript the authors show that deletion of Nrdp1 leads to formation of the round-headed sperm and male infertility by disrupting autophagy. It seems that Nrdp1 deficiency leads to male infertility probably because of impaired spermatogenesis after spermatid elongation. Furthermore, Nrdp1 deletion markedly impairs development of acrosomes in late phases of acrosome formation. TMT-labeled quantitative proteome analysis was used to determine up-regulated and down-regulated proteins in Nrdp1-deficient testis haploid cells in comparison with control samples from mice with heterozygous deletion of Nrdp1. According to the authors, deletion of Nrdp1 led to more proteins to be up-regulated in both testis and epididymis samples. This has to do with the role of Nrdp1 in ubiquitination, which regulates protein degradation through both proteasomes and lysosomes. Immunoblotting revealed that deletion of Nrdp1 reduces autophagic marker LC3-II, the lysosomal membrane protein LAMP1, the vesicle fusion protein VAMP8 and the autophagic cargo receptor OPTN. On the other hand, it increases the levels of the autophagic substrate p62 in mouse testes, suggesting that autophagy is generally inhibited in the Nrdp1-deficient testes. The authors conclude that Nrdp1 is required for the programmed autophagy, acrosome biogenesis, and proper mitochondrial function during spermiogenesis. Nrdp1 promotes Parkin ubiquitination, and regulates intracellular activity of Parkin, which promotes ubiquitination and degradation of the mitochondrial fission protein Drp1. Nrdp1 regulates mitophagy for mitochondrial quality control. Authors conclude that Nrdp1 is essential for the programmed acrosome formation, the proper sperm mitochondrial arrangement, sperm motility and male fertility, apparently by promoting autophagy during spermiogenesis.

Questions:

1. Nrdp1 is a RING-finger ubiquitin ligases possessing a large number of substrates. What are the main substrates for spermiogenesis accumulating in Nrdp1-deficient mice?

2. Nrdp1 is supposed to regulate protein degradation through both proteasomes and lysosomes. For proteasomal degradation it needs poly-ubiquitination via Lys48-chains. For autophagy and lysosomal degradation, it mostly requires mono-ubiquitination. How is this regulated? Is Nrdp1 part of a protein complex that regulates its specific activities?

Reviewer 2 Report

Spermatogenesis is the process by which haploid spermatozoa are produced from primordial germ cells, and the spermiogenesis is its final stage. During spermiogenesis the spermatids undergo toward a maturation process that implies: a) nuclear DNA condensation coordinated by the replacement of histones by protamines; b) formation of the sperm tail  connected with a mitochondria-rich midpiece, and c) development of the acrosome in the anterior part of the sperm nucleus. For the latter process, as well as for the sperm mitochondrial subcellular organization and localization, the autophagy represents a critical process. In the manuscript titled "Ubiquitin ligase Nrdp1 controls autophagy-associated acrosome biogenesis and mitochondrial arrangement during spermiogenesis" Luo Z-Y and colleagues report that the global deletion of the Nrdp1/RNF41 leads to mice male infertility due to formation of round-headed sperms because of an impairment in the autophagic flux. Hence, Nrdp1/RHF41 is a newly identified player required for spermatids  maturation. Further molecular characterization of either spermatids or sperm from Ndrp1 homozygous null mice, unveils that the amount of several proteins associated with mitochondria, acrosome and lysosomes were aberrantly accumulated, when compared to the wild type (w.t.) mice. Overall, this is the first report which associates the E3 Ubiquitin ligase Ndrp1/RNF41 to spermatogenesis/spermiogenesis. Nonetheless the evidence collected by the authors need to be implemented to fully support their conclusions. Indeed, their findings are, to some extent, quite indirect. Since autophagy relies mostly on ubiquitination, one would expect that autophagy inhibition leads to an accumulation of ubiquitinated protein. Hence, the authors should provide a Western blot with the ubiquitination pattern of the sperm from the Ndrp1 null mice compared with that of the w.t. The reviewer is aware that in vitro sperm culture is still an unmet need, thus the genetic rescue of the KO sperm is a hurdle. However, this can be easily bypassed by pharmacologically inhibiting, or promoting, the autophagy in sperm derived from w.t. and Nrdp1 homozygous null mice, by either of the following drugs, 3-methyladenine, chloroquine or ammonium chloride, rapamycin, metformin, etc... Eventually, while treatments with autophagy inhibiting drugs (3-methyladenine, chloquine, ammonium chloride) should mimic the effects caused by Nrdp1 deletion, those with autophagy promoting agents (rapamycin, metformin) should rescue the phenotype in terms of mitochondria subcellular localization, acrosome maturation etc..., and thus validating the conclusions. Fig. 1C: the authors are kindly invited to implement the data with the missing conditions (e.g. Nrdp1 w.t. male crossed either with the homozygous Nrdp1 null and heterozygous Nrdp1 female, respectively). Figures 2D and 3B: please implement the legend explaining what do the red arrows indicate, as it has been done in the case of asterisks. Fig. 3A: if possible improve the quality of the phase contrast panel for the w.t. Figures 7A and 7B: the quality of the Western blots is quite poor. Hence it is warmly advised to provide sharper pictures for the Nrdp1, VAMP8, LAMP1, p62, actin (in this specific case just a longer exposure is enough), LC3 and SIP. Quite a few typos are scattered throughout the manuscript (e.g. Fig. 2B Y axis labeling, I guess that it is not Motive sperm but Motile sperm, is it not?) Acronyms need to be detailed within the main text when cited for the first time (e.g. TMT, H & E, etc...).

The language is quite fine and requires minor editing.

Round 2

Reviewer 2 Report

The reviewer thanks very much the authors for the revised version of the manuscript. The concerns raised have partly addressed. However, quite few significant gaps remain to be filled. Indeed, the controls asked (Fig. 1C) are still missing. When it comes the Western Blots (Fig. 7A and 7B) the authors argue that they "...provided cleared results...", though in the version I have got such clearer quality improvement is not appreciable. It looks the same image displayed in the original version simply slightly adjusted. Eventually, the ubiquitination pattern might looks like that due to a couple of technical concerns that are below shortly discussed, and that should be carefully considered. Since ubiquitination is a reversible process at the time of cell lysis it might help to implement the lysis buffer by adding alkylating agents (e.g. iodoacetamide or N-ethylenmaleimide). The latter alkylate the active site cysteine residues of DUB enzymes, and thus preserving the ubiquitination status by inhibiting the Ub moyeties removal by DUBs. Second, Ubiquitin is a small globular protein very difficult to be denatured. Consistently, the ubiquitin epitopes might not be accessible to antibodies due to insufficient denaturation during SDS-PAGE. To bypass such hurdle and significantly enhances the signal it is recommended to denature the PVDF membrane before blocking it. The details regarding the anti-Ub antibody used (Section "Material and Methods") are missing.

The English language looks quite fine. Very minor issues are detected.

Round 3

Reviewer 2 Report

The author's claims are all understandable and to many extents reasonable. Nonetheless, confidence in data, that is based on proper controls, is even stronger than just belief.